# Acute kidney injury and its progression in hospitalized patients—Results from a retrospective multicentre cohort study with a digital decision support system

**Thea Sophie Kister** [1], **Johannes Remmler** [1], **Maria Schmidt** [1], **Martin Federbusch** [1], **Felix Eckelt** [1], **Berend Isermann** [1], **Heike Richter** [2], **Markus Wehner** [2], **Uwe Krause** [2], **Jan Halbritter** [3], **Carina Cundius** [4], **Markus Voigt** [4], **Alexander Kehrer** [5], **Jörg Michael Telle** [5], **Thorsten Kaiser** [1] *

1 Institute of Laboratory Medicine, Clinical Chemistry and Molecular Diagnostics (ILM), University of Leipzig Medical Center, Leipzig, Germany, 2 Muldentalkliniken GmbH Non-Profit Company, Hospital Grimma and Wurzen, Grimma, Germany, 3 Medical Department III, Division of Nephrology, University of Leipzig Medical Center, Leipzig, Germany, 4 Bereich 1 –Informationsmanagement, University of Leipzig Medical Center, Leipzig, Germany, 5 Xantas AG, Leipzig, Germany

☯ These authors contributed equally to this work.
* Thorsten.Kaiser@medizin.uni-leipzig.de

**Data Availability Statement:** Our data are available upon request. Although data are pseudonymised, details such as the combination of sex, age,

## Abstract

In this retrospective multicentric cohort study, we evaluate the potential benefits of a clinical decision support system (CDSS) for the automated detection of Acute kidney injury (AKI). A total of 80,389 cases, hospitalized from 2017 to 2019 at a tertiary care hospital (University of Leipzig Medical Center (ULMC)) and two primary care hospitals (Muldentalkliniken (MTL)) in Germany, were enrolled. AKI was defined and staged according to the Kidney disease: improving global outcomes (KDIGO) guidelines. Clinical and laboratory data was automatically collected from electronic patient records using the frameworks of the CDSS. In our cohort, we found an overall AKI incidence proportion of 12.1%. We identified 6,393/1,703/1,604 cases as AKI stage 1/2/3 (8.0%/2.1%/2.0%, respectively). Administrative coding with N17 (ICD-10-GM) was missing in 55.8% of all AKI cases with the potential for additional diagnosis related groups (DRG) reimbursement of 1,204,200 € in our study. AKI was associated with higher hospital mortality, increased length of hospitalisation and more frequent need of renal replacement therapy. A total of 19.1% of AKI cases (n = 1,848) showed progression to higher AKI stages (progressive AKI) during hospitalization. These cases presented with considerably longer hospitalization, higher rates of renal replacement therapy and increased mortality (p<0.001, respectively). Furthermore, progressive AKI was significantly associated with sepsis, shock, liver cirrhosis, myocardial infarction, and cardiac insufficiency. AKI, and especially its progression during hospitalization, is strongly associated with adverse outcomes. Our automated CDSS enables timely detection and bears potential to improve AKI outcomes, notably in cases of progressive AKI.

diagnoses, and period of assessment at ULMC and MTL could potentially be used to identify individual cases and patients. Therefore, in accordance with the General Data Protection Regulation, we are restrained regarding the public release of our data set. When administrative and legal requirements are met, trusted research institutions may request data access through the current director of the Institute of Laboratory Medicine, Clinical Chemistry and Molecular Diagnostics Leipzig (MB-sek-ilm@medizin.uni-leipzig.de) or the AMPEL project directly (MB-ilm-ampel@medizin.uni-leipzig.de).

**Funding:** We acknowledge support from Leipzig University for Open Access Publishing. This study is a part of the AMPEL project, which is co-financed through public funds according to the budget decided by the Saxon State Parliament (RL eHealthSax 2017/18, grant No.100331796, URL: https://www.sachsen.de). The funder provided support in the form of salaries for authors MS and FE, but did not have any additional role in the study design, data collection and analysis, decision to publish, or preparation of the manuscript. The specific roles of these authors are articulated in the 'author contributions' section. Coauthors JMT and AK are employees of Xantas Ag. The company receives funding by RL eHealthSax 2017/18 for the sole purpose of technically implementing a CDSS, which they may use commercially. Yet, they did not participate in the preparation of the frameworks, study design, data collection and analysis or the decision to publish. During the preparation of the manuscript, they contributed to the development of the software, review and editing. The specific roles of these authors are articulated in the 'author contributions' section.

**Competing interests:** The authors have declared that no competing interests exist. The affiliation with Xantas AG is not commercial. AK and JMT are employees of Xantas AG. The company receives funding by RL eHealthSax 2017/18 for the sole purpose of technically implementing a CDSS, which they may use commercially. Yet, they did not participate in the preparation of the frameworks, study design, data collection and analysis or the decision to publish. There are no patents, products in development or marketed products associated with this research to declare. This does not alter our adherence to PLOS ONE policies on sharing data and materials.

# Introduction

Acute kidney injury (AKI) is a common and serious clinical event that affects up to 15.0% of all hospitalized and up to 50.0% intensive care unit patients [1]. In studies using definitions conforming to the Kidney disease: improving global outcomes criteria (KDIGO), the pooled rate of AKI was 23.2% worldwide, and the AKI-associated mortality was 23.0% [2]. AKI is associated with an increased short- and long-term mortality, as well as the development of chronic kidney disease (CKD) [3]. Pre-existing CKD has been identified as a risk factor for AKI associated with increased mortality, in part due to the increased risk of AKI-associated end-stage renal disease [4]. According to the KDIGO Clinical Practice Guidelines, AKI is defined by an increased serum creatinine (SCr) and/or reduced urine output, separated into three different stages of severity (Table 1) [5]. Even an increase of only 26.5 μmol/l in SCr (AKI stage 1) is correlated with a significant risk of mortality and morbidity [6], requiring determination of the underlying cause in a timely manner. Patients should be managed according to their susceptibilities and an individualized therapy based on patient risk has to be conducted [5, 7]. However, due to the complexity of the AKI definition, diagnosis could be delayed or even overlooked.

## Study aims and objectives

For this reason, we developed an automated diagnostic system that identifies patients with AKI immediately after laboratory diagnostics as a part of a digital clinical decision support system (CDSS). To evaluate the system, we applied it retrospectively to all hospitalized patients during the years 2017 to 2019 and analysed the incidence proportion of AKI and the distribution of the three stages at a university hospital and two regional hospitals. Thus, we analysed differences in AKI reporting between a tertiary care hospital and primary care institutions. Furthermore, administrative coding [8] and the involvement of nephrologists [9] in AKI diagnosis was investigated. In particular, cases with further decrease in kidney function, and therefore, a progression to a higher AKI stage of severity during hospitalization (progressive AKI) were analysed in more detail.

# Materials and methods

## Study population

In this retrospective observational cohort study, all hospitalized cases (n = 227 194) admitted between 1st January 2017 and 31st December 2019 at the University of Leipzig Medical Center (ULMC, 1 451 beds) and the Muldentalkliniken (MTL, hospital locations Grimma (177 beds) and Wurzen (178 beds)) were analysed, including repeated hospital admissions. The ULMC is

**Table 1. AKI staging according to the KDIGO guidelines.**

| Stage | Serum creatinine | Urine output |
|---|---|---|
| 1 | 1.5–1.9 times baseline, OR | <0.5 ml/kg/h for 6–12 hours |
| | ≥0.3 mg/dl (≥26.5 μmol/l) increase | |
| 2 | 2.0–2.9 times baseline | <0.5 ml/kg/h for ≥12 hours |
| 3 | 3.0 times baseline, OR | <0.3 ml/kg/h for ≥24 hours, OR |
| | Increase in serum creatinine to ≥4.0 mg/dl (≥353.6μmol/L), OR | Anuria for ≥12 hours |
| | Initiation of renal replacement therapy, OR | |
| | In patients <18 years, decrease in eGFR to <35 ml/min per 1.73 m$^2$ | |

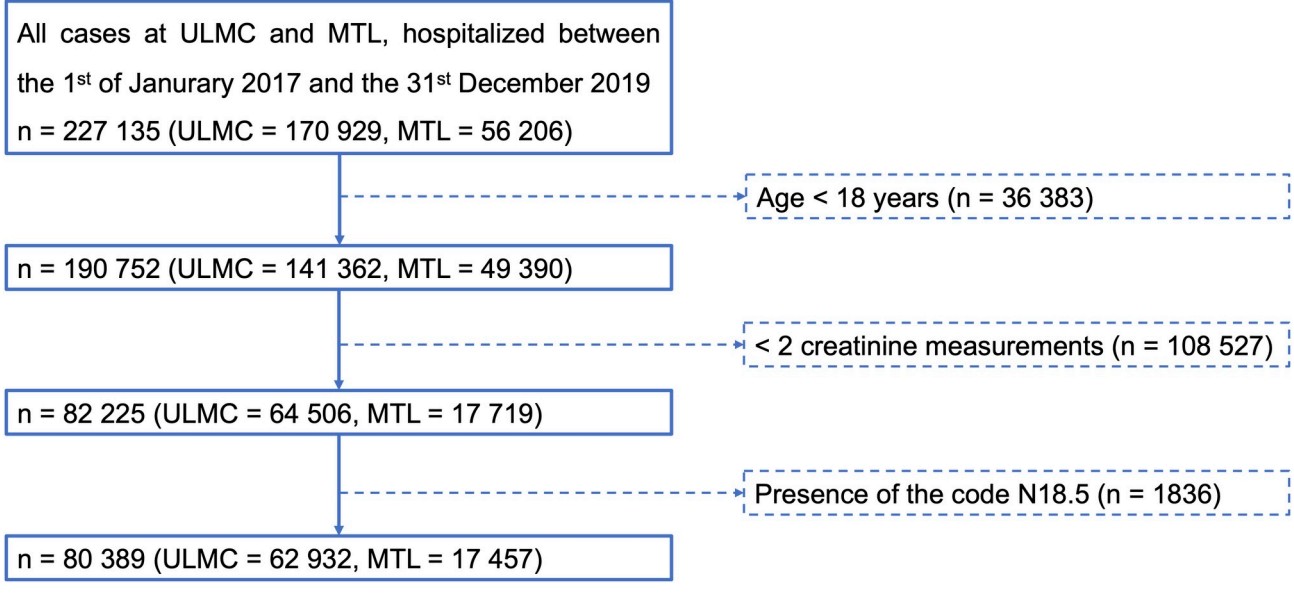

**Fig 1. Study cohort with inclusion and exclusion criteria.**

a tertiary care hospital with its own department of nephrology, several internists with specialization in nephrology and an inpatient dialysis department.

MTL hospitals are primary health care institutions without a nephrology department or internists specialized in nephrology. Instead, they cooperate with an outpatient dialysis centre (KfH, board of trustees for dialysis and kidney transplantation) in Grimma. The study population consisted of all adult patients ($\geq$18 years) (n = 190 752 cases) who received at least two creatinine measurements (n = 82 225 cases). Patients with diagnosis coded according the International Statistical Classification of Diseases and Related Health Problems–German Modification (ICD-GM) N18.5 (dialysis-dependent chronic kidney disease) [10] were excluded from our study (Fig 1, n = 80 389 cases remaining as study cohort). Cases with kidney transplantation in patients' history were not excluded unless the diagnosis "N18.5 (dialysis-dependent chronic kidney disease)" was also coded. This study has been approved by the Ethics Committee of the Medical Faculty of the University of Leipzig, Germany (No. 214/18-ek). AKI was defined and staged according to the KDIGO Clinical Practice Guideline for Acute Kidney Injury (Table 1) [5].

### Acquisition of data

Clinical and laboratory data was automatically collected from electronic patient records within the clinic information system using the frameworks of the AMPEL system [11] (www.ampel. care), a CDSS, to monitor laboratory results. AMPEL was recently implemented at ULMC and MTL. Within the system, follow-up treatments and examinations are monitored, and patients' caregivers are notified when life threatening conditions have been detected (Walter Costa MB et al. The Clinical Decision Support System AMPEL for Laboratory Diagnostics: Implementation and Technical Evaluation. Journal of Medical Internet Research, minor revision). Extracted data included age, sex and creatinine measurements (together with estimated glomerular filtration rates (GFR)). Additionally, relevant diagnoses/comorbidities [1], which are coded based on ICD-10-GM [10], and case outcomes (lengths of hospitalization, hospital mortality, eGFR on discharge, presence of nephrological consultations, at least one renal

replacement therapy or renal replacement therapy within 72 hours of discharge) were retrieved. We used a second data source (the laboratory information system) to independently validate the automated AKI detection by the AMPEL system. All data were pseudonymized for analysis. The retrospective use of medical records was approved by the ethics committee and works according to an opt-out system. Hence, no additional informed consent was required.

AKI was defined and staged based on absolute and relative changes of creatinine levels within the case, according to KDIGO-criteria, with the lowest sCr level of the last 168 hours serving as a baseline in this definition (Table 1). Urine excretion as an AKI criterion was not available. All creatinine measurements were performed in serum on a Cobas 8000 Analyzer (Roche, Mannheim, Germany; Creatinine Plus Ver. 2 kit, enzymatic method). The highest measured AKI stage determined the classification into the three severities. The progression of AKI was classified according to the first and the maximum determined AKI stage. eGFR was estimated using the Chronic Kidney Disease Epidemiology Collaboration equation (CKD-EPI) [12] without consideration of ethnicity.

## Statistics

A case-based analysis was performed using Microsoft Excel for Office 365 ProPlus (Microsoft Corporation, Redmond, USA). Further statistical analysis was performed using SPSS 25 (SPSS Inc., Chicago, USA), MedCalc 12 (MedCalc Software bvba, Ostend, Belgium) and R 4.0.2.

The characteristics of the cohort are shown as median and interquartile ranges (IQR). Mann-Whitney U test was used to compare continuous characteristics between two groups, and Pearson's chi-squared test was used to compare categorical data.

## Results

### Study cohort characteristics and differences between primary and tertiary care centres

Between 1st January 2017 and 31st December 2019, 80 389 (62 932 at ULMC and 17 457 at MTL) hospitalized cases met the inclusion criteria (Fig 1). In our cohort we found a creatinine-based AKI incidence proportion of 12.1% (9 700 of 80 389 cases), which was higher in the tertiary care hospital (ULMC) (12.4%) than in MTL (10.9%) (p<0.001) (Table 2).

**Table 2. Study cohort and a comparison of the incidence proportion, the basic patient characteristics, and the case outcomes between ULMC and MTL.**

| | All | ULMC (tertiary health care institution) | MTL (primary health care institution) |
|---|---|---|---|
| **Basic patient characteristics** | | | |
| Age, median [IQR] *** | 67 [55–79] | 65 [52.5–77.5] | 78 [69–87] |
| Sex, % male *** | 54.7 | 56.7 | 47.5 |
| First eGFR, median [IQR] *** | 73 [50–96] | 76 [54–98] | 49 [30–68] |
| **Case outcomes** | | | |
| eGFR on discharge#, median [IQR] *** | 79 [58–100] | 81 [61–101] | 58 [38–78] |
| Total length of hospitalization (days), median [IQR] *** | 8 [3.5–12.5] | 8 [3–13] | 7 [4–10] |
| Hospital mortality, % | 4.8 | 4.8 | 5.0 |
| Dialysis at least once, % *** | 2.4 | 2.8 | 0.8 |
| Dialysis within 72 hours of discharge, % *** | 1.3 | 1.5 | 0.3 |
| **Incidence proportion** | | | |
| AKI overall hospital incidence, n, % *** | 9700, 12.1 | 7800, 12.4 | 1900, 10.9 |

Asterisks indicate significant differences between ULMC and MTL (p-value *<0.05, **<0.01, ***<0.001).

#calculated from all patients who left the hospital alive and without dialysis within 72 h of discharge, given in ml/min/1.73m$^2$ (CKD-EPI formula)

Considering the basic case characteristics, we observed significant differences in the median age: ULMC: 65, MTL: 78 (p<0.001). Women were underrepresented at ULMC, but overrepresented at MTL. Regarding the case outcomes, patients were hospitalized for a median of 8 days at ULMC and 7 days at MTL (p<0.001). The overall hospital mortality was 4.8% without significant differences between the two types of hospitals (p = 0.359). We observed a lower initial eGFR in cases at MTL (49 ml/min), compared to cases at ULMC (76 ml/min) (p<0.001). The frequency of need for at least one renal replacement therapy differed significantly between ULMC (2.8%) and MTL (0.8%) (p<0.001).

## Comparison of AKI cases with non-AKI cases and in-between AKI stages

We identified 6 393 cases as AKI stage 1 (AKIN1) (8.0%), 1 703 as AKI stage 2 (AKIN2) (2.1%) and 1 604 as AKI stage 3 (AKIN3) (2.0%) (Table 3). Median patient age of AKI cases was 5 years higher than that of non-AKI cases, whereas among AKI cases, higher stages were associated with younger age. eGFR at discharge did not differ significantly between AKIN1 and AKIN2 cases (p = 0.147), although it was lower in AKIN3 cases (p<0.001). The frequency of renal replacement therapy (at least one dialysis as well as dialysis within 72 hours of discharge) also increased with the AKI stages (Table 3). Additionally, the median length of hospitalization, as well as the hospital mortality, rose with AKI stage (Figs 2 and 3). Notably, in comparison to non-AKI cases, the length of hospitalization, the hospital mortality and need of renal replacement therapy were higher in AKI cases, whereas the eGFRs (first and before discharge)

**Table 3. Patient characteristics and comparison of the incidence, case outcomes and comorbidities among the AKI stages.**

|  | No AKI | AKIN1 | AKIN2 | AKIN3 |
|---|---|---|---|---|
| Incidence proportion, n, % | 70689, 87.9 | 6393, 8.0 | 1703, 2.1 | 1604, 2.0 |
| **Basic Patient characteristics** | | | | |
| Age, median [IQR] *** | 67 [55–79] | 73 [63–83] | 70 [59.5–80.5] | 69 [59–79] |
| Sex, % male *** | 54.4 | 56.2 | 55.2 | 64.3 |
| First eGFR, median [IQR] *** | 75 [53–97] | 56 [32–80] | 63 [40–86] | 42 [10.5–73.5] |
| **Case outcomes** | | | | |
| eGFR on discharge#, median [IQR] *** | 81 [60.5–101.5] | 55 [32.5–77.5] | 62 [37–87] | 48 [19.5–76.5] |
| Total length of hospitalization (days), median [IQR] *** | 7 [3–11] | 14 [6–22] | 19 [7.5–30.5] | 20 [7–33] |
| Hospital mortality, % *** | 2.3 | 15.6 | 34.2 | 41.4 |
| Dialysis at least once, % *** | 0.5 | 7.8 | 22.0 | 44.0 |
| Dialysis within 72 hours of discharge, % *** | 0.3 | 3.9 | 10.7 | 21.0 |
| Intensive or intermediate care, % *** | 13.1 | 26.6 | 29.7 | 32.3 |
| **Comorbidities** | | | | |
| I10.- Hypertension, % *** | 46.3 | 49.1 | 50.3 | 46.8 |
| E11.- Diabetes mellitus, % *** | 25.5 | 36.1 | 34.6 | 37.5 |
| E86.- Exsiccosis, % *** | 5.0 | 6.3 | 6.9 | 8.7 |
| R57.- Shock, % *** | 1.3 | 12.2 | 27.5 | 31.7 |
| I25.- Coronary heart disease, % *** | 13.9 | 21.4 | 19.6 | 20.8 |
| I21.- Myocardial infarction, % *** | 1.8 | 3.8 | 4.1 | 4.3 |
| I50.- Cardiac insufficiency, % *** | 15.1 | 32.3 | 32.9 | 37.4 |
| A41.- Sepsis, % *** | 3.0 | 16.0 | 30.5 | 35.0 |
| K74.- Liver cirrhosis, % *** | 1.8 | 3.7 | 6.9 | 5.8 |

Asterisks indicate significant differences between AKI-cases and No-AKI-Cases (p-value *<0.05, **<0.01, ***<0.001)

#calculated from all patients who left the hospital alive and without dialysis within 72 h of discharge, given in ml/min/1.73m$^2$ (CKD-EPI formula)

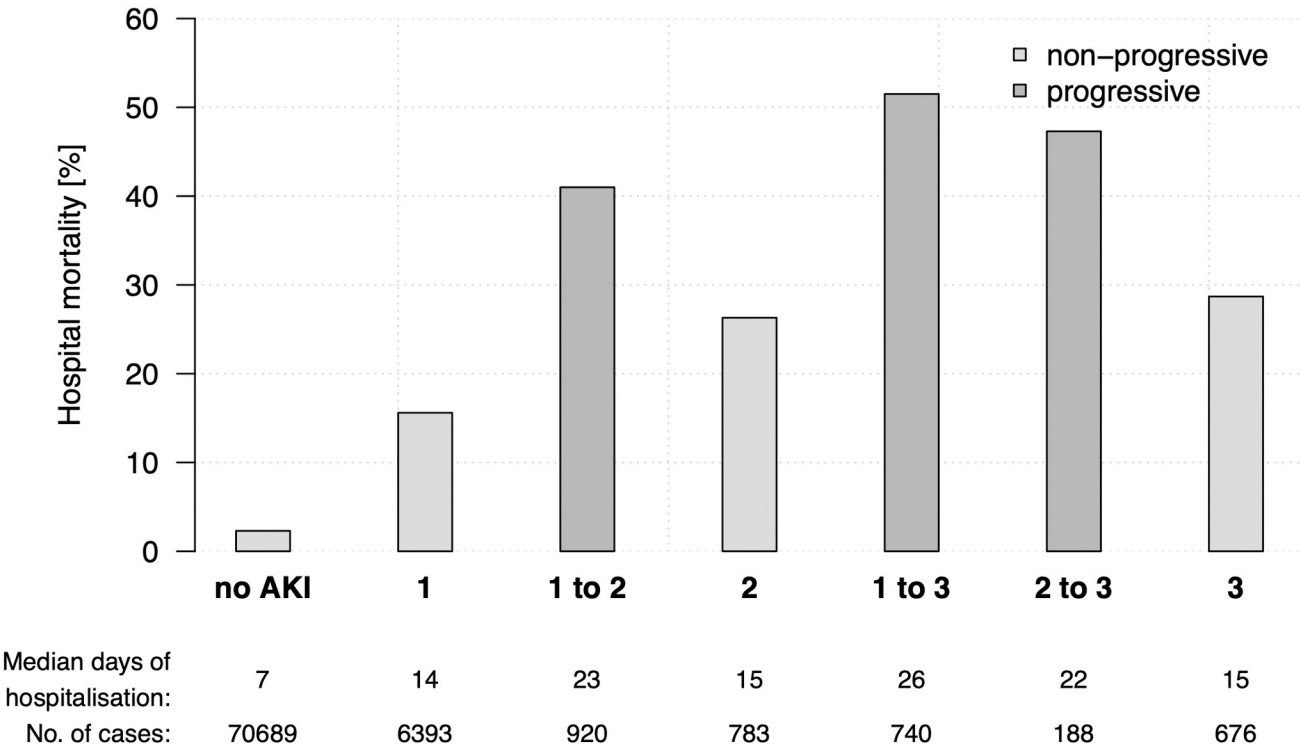

**Fig 2. Mortality of patients with progressive and non-progressive AKI, and those without AKI.** Hospital mortality in % for progressive and non-progressive AKI. Patients were divided into six groups according to the first and maximum detected AKI stage, which is equal for non-progressive AKI (AKIN1, AKIN2, AKIN3) and given as AKIN X to Y (X = first and Y = max AKI stage) for progressive AKI.

were lower (Table 3). We observed associations between selected comorbidities and AKI. The incidence proportions of hypertension, diabetes mellitus, exsiccosis, shock, coronary heart disease, myocardial infarction, cardiac insufficiency, sepsis, and liver cirrhosis were higher in cases with AKI than in non-AKI cases.

## Administrative coding of AKI

Overall, we noted a high absence of administrative coding as N17 (ICD-10-GM code for Acute kidney failure) in AKIN1 (68.1%), AKIN2 (41.0%) and AKIN3 (22.3%) cases (Table 4). Yet, almost every AKI was followed up with at least one creatinine measurement after its detection (ULMC/MTL in 88.4%/81.1% of the cases, respectively). The frequency of follow-up and coding increased with higher AKI stages. In 5.3% of all non-AKI cases, N17 was coded. Considering the German diagnosis and procedure related reimbursement system, additional revenues of 1 204 200 € could have been generated within the three years by correct administrative coding of the cases within our cohort.

## Progressive AKI and its clinical implications

In a second step, we compared cases with progressive AKI (AKIN1→AKIN2, AKIN1→AKIN3, AKIN2→AKIN3) to those without disease progression during hospitalization (AKIN1→AKIN1, AKIN2→AKIN2, AKIN3→AKIN3, first→maximum AKI) (Table 5 and S1 Table). Although progressive AKI was evident in 1,848 cases (19.1% of all AKI cases), these were likely to be younger in median age (p<0.001) and predominantly male (p = 0.006). As progressive AKI was associated with longer hospitalisation and higher rates of renal

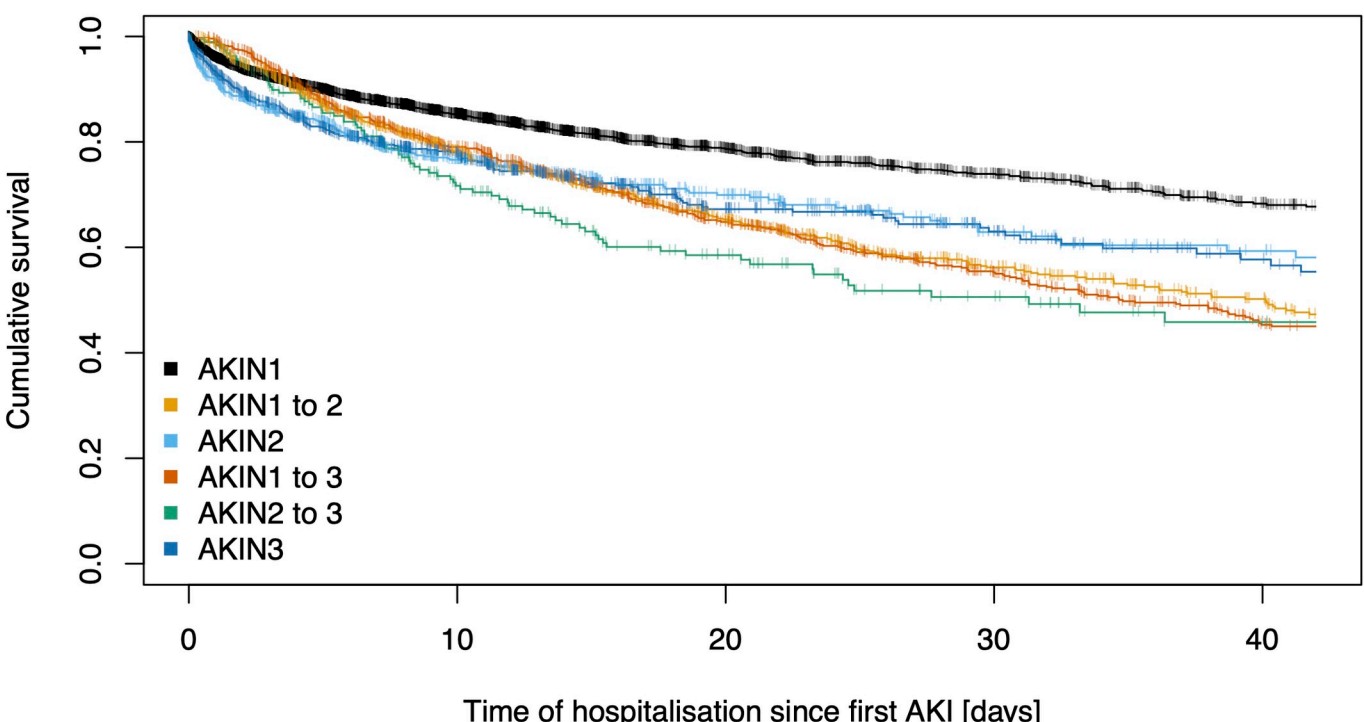

**Fig 3. Survival of patients with progressive and non-progressive AKI.** Kaplan-Meier curves for the first 42 days of hospitalisation after first AKI detection. Patients were divided into six groups according to first and maximum detected AKI stage, which is equal for non-progressive AKI (AKIN1, AKIN2, AKIN3) and given as AKIN X to Y (X = first and Y = max AKI stage) for progressive AKI. Censored survival data due to hospital discharge are indicated by +. Median lengths of hospitalisation were 14, 23, 15, 26, 22, 15 days for AKIN1, AKIN1 to 2, AKIN2, AKIN1 to 3, AKIN2 to 3 and AKIN3, respectively.

replacement therapy and mortality, the overall outcome of patients with progressive AKI was worse, compared to cases without progression. Especially with regard to mortality, progressive AKI seems to be a significant prognosis factor (Figs 2 and 3). This is also reflected in higher rate of intermediate or intensive care among cases with progressive AKI and reflects also in higher administrative N17 coding. In a minority of cases, completely documented nephrological consultations occurred and were more often conducted in cases with progression (analysis for ULMC only). Cases with progressive AKI showed significantly higher incidence proportions for certain comorbidities; we observed a strong association with sepsis, shock, liver cirrhosis, myocardial infarction, and cardiac insufficiency (Table 5 and S1 Table).

## Discussion

In this study, we confirmed a high frequency of AKI among our cohort and demonstrated its association with severe comorbidities and increased mortality during hospitalization. In line

**Table 4. Comparison of the administrative coding among the AKI stages.**

| | Administrative coding | | | | |
|---|---|---|---|---|---|
| | **No AKI** | **AKIN1** | **AKIN2** | **AKIN3** | **All AKI** |
| Only N17 (Acute kidney injury), % | 2.1 | 14.4 | 36.1 | 42.5 | 22.9 |
| Only N18 (Chronic kidney disease), % | 23.0 | 24.3 | 10.0 | 8.9 | 19.2 |
| N17 + N18, % | 3.1 | 17.5 | 22.9 | 35.2 | 21.3 |
| Neither N17 nor N18, % | 71.7 | 43.8 | 31,1 | 13.5 | 36.6 |

**Table 5. Comparison of cases with progressive and non-progressive AKI cases.**

|  | Cases with progressive AKI | Cases without progressive AKI |
|---|---|---|
| Frequency, n, % | 1848, 19.1 | 7852, 80.9 |
| **Basic patient characteristics** |  |  |
| Age (years), median [IQR] *** | 69 [58.5–79.5] | 73 [63–83] |
| Sex, % male ** | 60.2 | 56.7 |
| First eGFR, median [IQR] | 55 [29.5–80.5] | 57 [32–82] |
| **Administrative coding** |  |  |
| N17 coded, % *** | 76.7 | 36.6 |
| **Case outcomes** |  |  |
| eGFR on discharge#, median [IQR] *** | 55 [30–80] | 77 [53–101] |
| Total length of hospitalization (days), median [IQR] *** | 24 [14–34] | 14 [4.5–3.5] |
| Hospital mortality, % *** | 45.8 | 17.8 |
| Dialysis at least once, % *** | 41.5 | 10.4 |
| Dialysis within 72 hours of discharge, % *** | 18.9 | 5.3 |
| Intensive or intermediate care, % *** | 37.0 | 26.0 |
| **Nephrological consultation and time interval** |  |  |
| Nephrological consultation at least once (ULMC only), % *** | 6.1 | 2.2 |
| Time from first AKI to first nephrological consultation (ULMC only), h, median | 100.3 | 52.3 |
| Time interval between first AKI and AKI max, h, median | 56.3 | - |
| **Comorbidities** |  |  |
| Hypertension—I10., % | 46.9 | 49.4 |
| Diabetes mellitus—E11., % | 36.0 | 36.1 |
| Liver cirrhosis—K74., % *** | 8.4 | 3.7 |
| Coronary heart disease—I25., % | 20.7 | 21.1 |
| Myocardial infarction—I21., % *** | 5.0 | 3.7 |
| Cardiac insufficiency—I50., % *** | 38.5 | 32.0 |
| Exsiccosis—E86., % | 6.1 | 7.0 |
| Shock—R57., % *** | 39.3 | 13.1 |
| Sepsis—A41., % *** | 40.0 | 17.3 |

Asterisks indicate significant differences between cases with progressive AKI and cases without AKI progression (p-value *$<0.05$, **$<0.01$, ***$<0.001$).

#calculated from all patients who left the hospital alive and without dialysis within 72 h of discharge, given in ml/min/1.73m$^2$ (CKD-EPI formula)

with previous studies, we reported a creatinine-based AKI incidence proportion of 12.1%. Studies published after the implementation of the KDIGO guidelines in 2012 have found incidences ranging from 10.7% [2], 15.3% [13], 18.3% [14], 22.7% [15], 23.2% [2] to 31.3% [8].

Despite its high incidence, AKI is still an underrated disease. In our study, we observed low rates of administrative coding, a phenomenon which was also observed in other studies [8, 16–19]. This may indicate insufficient consideration or detection of the disease, which could contribute to worse outcomes, given that documentation/coding is associated with improved outcomes after adjustment for AKI severity [16]. Additionally, the financial aspect of a correct and complete coding is evident, considering the German diagnosis and procedure related reimbursement system.

To the best of our knowledge, our study is the first comparative analysis of AKI between a tertiary care hospital and two primary care institutions in Germany, and we found comparable

AKI incidences proportions. However, patients in these sites differed significantly in age and sex. Age differences are easily explained by rural and urban demographics within the catchment area [20]. This may also partly explain the lower eGFRs at admission and discharge at MTL [21]. Additionally, stricter indication for follow-up measurements at MTL could have caused a selection bias towards patients in worse conditions, as a second creatinine measurement (part of our inclusion criteria) was performed more frequent at ULMC (45.2% of all hospitalized cases, compared to 35.8% at MTL). At ULMC, the rates of renal replacement therapy were more than twice as high compared to those at MTL, which is most likely due to a missing department of nephrology and in-house dialysis centre at MTL, where critically-ill patients with a renal disease are transferred to another hospital in most cases. Interestingly, the hospital mortality within our cohort did not differ significantly between the two sites.

The presence, as well as the stage of AKI, was strongly associated with the case outcomes in our cohort. Thereby, our results confirmed those of previous studies showing that cases with AKI had longer lengths of hospitalization, higher rates of renal replacement therapy and a higher risk of admission to intensive care units as well as higher hospital mortality than cases without AKI [13]. These outcome measures worsened with increasing AKI stage, in accordance with the literature [18, 22]. Of note, in our study, are the substantial differences between non-AKI and AKIN1 cases, matching findings that even modest changes in SCr are associated with higher mortality [6] (Table 3). AKI is associated with several comorbidities across several medical conditions, and may represent an additional risk factor in affected patients. Hypertension [23], diabetes mellitus [24], volume depletion such as exsiccosis and shock [25], cardiological complications (e.g. coronary heart disease, myocardial infarction and cardiac insufficiency [26]), sepsis [27] and liver cirrhosis [28] were often discovered alongside AKI. In our cohort, we confirmed, in particular, a strong association of AKI with sepsis and shock (Table 3). Regarding eGFRs (first eGFR as well as eGFR at discharge), we observed higher values in AKIN2 compared to AKIN1 (Table 3). This unexpected finding could be explained by the limitation of the KDIGO criteria: AKIN1 is defined combinatorically by a relative (SCr > 50% over 7 days) and an absolute (SCr > 0.3mg/dL over 48h) criterion for SCr dynamics [29], whereas for AKIN2, only the relative criterion (SCr > 100% over 7 days) is relevant (Table 1). The absolute criterion is reached faster in patients with high baseline SCr. Similarly, the overrepresentation of male sex among our AKI cases might have presented another limitation to the absolute criterion (SCr > 0.3mg/dL over 48h): lower muscle mass and associated lower SCr levels could underestimate the degree of renal failure in women and elderly patients [30].

To the best of our knowledge, our study is the first to distinguish between AKI cases with and without progression to a higher stage during hospitalization, with respect to characteristics and outcomes of these cases. In our cohort, 19.1% of all AKI cases showed a progression to a higher stage. For these patients, we discovered considerable worse outcomes, especially concerning hospital mortality. Cases with a progressive AKI were also associated with higher rates of renal replacement therapy, longer lengths of hospitalization and more frequent intensive care treatments. Progressive AKI is strongly associated with the highly relevant comorbidities sepsis and shock, as well as liver cirrhosis, myocardial infarction, and cardiac insufficiency. A distinctive restriction of the circulation and, therefore, a disturbed perfusion as a common effect of these diseases could be assumed [31, 32]. Progressive AKI indicates a significantly worse prognosis. Thus, an immediate detection of any AKI and further medical work-up of the patient according to the KDIGO guideline seem urgently necessary and might prevent progressive AKI and further complications. However, due to the complicated criteria, timely identification of these patients is difficult in routine hospital care workflow. Our AMPEL-CDSS can identify these patients in a timely manner and notify clinicians immediately. Consequently, we are planning a prospective trial to verify that this CDSS can improve renal and

other clinical outcomes. In the process of diagnosis and therapy, documented nephrological consultations rarely occurred in our cohort. If consultations were ordered, they were often delayed. Balasubramanian *et al.* showed a significant risk reduction of further kidney function deterioration if renal consultations took place within 18 hours of AKI-onset [9, 33]. Delayed consultations in our cohort might have been due to missed diagnosis and a lack of consciousness of the significance of timely intervention [34] and could be improved by the AMPEL-CDSS as well.

Limitations of our study are its retrospective nature and the AKI definition solely based on serum creatinine dynamics. Urine excretion as an AKI criterion was not included, as urine excretion is often not systematically documented. This might cause an underestimation of the AKI incidence proportion, which is reflected by the fact that 5.3% of all non-AKI cases in our cohort were coded with N17. Furthermore, we exclusively included inpatient cases. Thus, pre-stationary diagnostics and long-term outcome data, such as outpatient deaths, were not available. An analysis of patient transfers from MTL to a tertiary care hospital was not performed.

Nevertheless, our automated approach was necessary to evaluate the potentials of the AMPEL-CDSS (which depends on these automated analysis of limited data) and enabled the investigation of a large cohort in a multicentric manner. Thus, we were able to comparatively analyse AKI cases on two different levels of medical care in Germany. Moreover, our study uniquely investigated and proved the substantial prognostic value of a progressive AKI during hospitalization. Our results show the potentials and importance of an automated, real-time detection of AKI and its progression, which might improve patient outcome [35]. This is the designated aim of our CDSS called AMPEL, which is currently being implemented and will be further evaluated in prospective studies.

## Supporting information

**S1 Table. Comparison of the frequency, the patient characteristics, the administrative coding, the case outcomes and the comorbidities between cases with progressive AKI and those without.**
(PDF)

## Author Contributions

**Conceptualization:** Thea Sophie Kister, Johannes Remmler, Thorsten Kaiser.

**Data curation:** Thea Sophie Kister, Johannes Remmler, Maria Schmidt.

**Formal analysis:** Thea Sophie Kister, Johannes Remmler, Maria Schmidt.

**Funding acquisition:** Heike Richter, Jörg Michael Telle, Thorsten Kaiser.

**Investigation:** Thea Sophie Kister, Johannes Remmler.

**Methodology:** Thea Sophie Kister, Johannes Remmler.

**Project administration:** Heike Richter, Jörg Michael Telle, Thorsten Kaiser.

**Resources:** Heike Richter, Thorsten Kaiser.

**Software:** Carina Cundius, Markus Voigt, Alexander Kehrer, Jörg Michael Telle, Thorsten Kaiser.

**Supervision:** Thorsten Kaiser.

**Validation:** Thea Sophie Kister, Johannes Remmler, Maria Schmidt.

**Visualization:** Thea Sophie Kister, Johannes Remmler, Maria Schmidt.

**Writing – original draft:** Thea Sophie Kister, Johannes Remmler.

**Writing – review & editing:** Maria Schmidt, Martin Federbusch, Felix Eckelt, Berend Isermann, Heike Richter, Markus Wehner, Uwe Krause, Jan Halbritter, Carina Cundius, Markus Voigt, Alexander Kehrer, Jörg Michael Telle, Thorsten Kaiser.

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
