## [Decision Letter · Decision Letter 0]

19 May 2021

PONE-D-21-07899

Acute Kidney Injury and its Progression in Hospitalized Patients - Results from a Retrospective Multicentre Cohort Study with a Digital Decision Support System

PLOS ONE

Dear Dr. Kister,

Thank you for submitting your manuscript to PLOS ONE. After careful consideration, we feel that it has merit but does not fully meet PLOS ONE’s publication criteria as it currently stands. Therefore, we invite you to submit a revised version of the manuscript that addresses the points raised during the review process.I would like this manuscript to address few of the changes as requested by the reviewer as it can considered for publication after various queries have been addressed.

We look forward to receiving your revised manuscript.

Kind regards,

Bhagwan Dass, MD

Academic Editor

PLOS ONE

Journal Requirements:

2)  Thank you for stating the following in the Competing Interests section:

[The authors have declared that no competing interests exist.].   

We note that one or more of the authors are employed by a commercial company: Xantas AG

i. Please provide an amended Funding Statement declaring this commercial affiliation, as well as a statement regarding the Role of Funders in your study. If the funding organization did not play a role in the study design, data collection and analysis, decision to publish, or preparation of the manuscript and only provided financial support in the form of authors' salaries and/or research materials, please review your statements relating to the author contributions, and ensure you have specifically and accurately indicated the role(s) that these authors had in your study. You can update author roles in the Author Contributions section of the online submission form.

ii. Please also provide an updated Competing Interests Statement declaring this commercial affiliation along with any other relevant declarations relating to employment, consultancy, patents, products in development, or marketed products, etc.  

3)  Please provide additional details regarding participant consent. In the ethics statement in the Methods and online submission information, please ensure that you have specified (1) whether consent was informed and (2) what type you obtained (for instance, written or verbal, and if verbal, how it was documented and witnessed). If your study included minors, state whether you obtained consent from parents or guardians. If the need for consent was waived by the ethics committee, please include this information.

Reviewers' comments:

Reviewer's Responses to Questions

**Comments to the Author**

1. Is the manuscript technically sound, and do the data support the conclusions?

Reviewer #1: Yes

Reviewer #2: Yes

2. Has the statistical analysis been performed appropriately and rigorously? 

Reviewer #1: I Don't Know

Reviewer #2: I Don't Know

3. Have the authors made all data underlying the findings in their manuscript fully available?

Reviewer #1: No

Reviewer #2: Yes

4. Is the manuscript presented in an intelligible fashion and written in standard English?

Reviewer #1: Yes

Reviewer #2: Yes

5. Review Comments to the Author

Reviewer #1: Please see my comments on different sections of the manuscript:

Introduction: The rationale for the study is outlined well. While it is more of a stylistic comment, it would be clearer to the reader if there was a separate paragraph for study aims and if they were enumerated.

Methods:

- Were kidney transplant recipients included or excluded from the study. This has not been stated.

- KDIGO classification / definition of AKI relies on knowing baseline serum creatinine. How were baseline serum creatinine levels ascertained?

- KDIGO classification also establishes AKI diagnosis based on urine output. This study does not include urine assessment. This is correctly stated in the limitation and should be made clear in the methods as well.

- What was done for the same patient’s multiple admissions? This is unclear. If there are repeat admissions (as they are bound to be in healthcare systems), are all admissions included or only the first admission included? This is not clear.

- It is also unclear if patients were transferred from primary hospitals to tertiary hospitals. If they are transferred, are they counted twice? Was this analyzed?

Results:

- It is not very surprising that tertiary care center was found to have a higher burden of AKI. From Table 2, it appears that MTL has older patients with lower baseline GFR. So, it is little surprising that despite having 11% AKI incidence, there were lower need for dialysis in MTL. This makes me wonder if patients were transferred to the tertiary hospital if dialysis was needed and that (if true) is not accounted for in the analysis or in the interpretation of data.

- As patients with AKI have a higher co-morbidity burden, it is unclear if AKI is an independent predictor of mortality or a marker of poor prognosis in this dataset (relating to Table 3). Multivariable analysis would be helpful. Dataset has enough event rate for multivariable analysis testing AKI as an independent predictor for mortality.

- Regarding the association between progressive AKI and clinical outcomes, does the site (ULMC vs. MTL) alone account for the differences in clinical outcomes related to age? ULMC patients were younger and, since it is a tertiary care center, likely sicker than MTL patients who were on average 13 year older. It would be helpful if this data was divided by sites (ULMC and MTL) to know if the relationship between progressive AKI and younger age still holds independently.

Supplementary information - I am not sure if Plos One would require underlying data behind summary statistics to be provided as a data table. Summary statistics are well highlighted well but I did not see the underlying data points behind summary statistics.

Reviewer #2: Authors have showed the utility of AMPLE-CDSS with progression of AKI during hospitalization in their retrospective data collection among the defined three hospitals. This study is in the direction of our understanding whether this tool will ultimately help in improving AKI and patient outcomes. Well executed and well summarized data collection.

6. PLOS authors have the option to publish the peer review history of their article (what does this mean?). If published, this will include your full peer review and any attached files.

Reviewer #1: **Yes: **Ashish Upadhyay MD

Reviewer #2: No

---

## [Author Response · Author response to Decision Letter 0]

23 Jun 2021

Comments to the Author

1. Is the manuscript technically sound, and do the data support the conclusions?

Reviewer #1: Yes

Reviewer #2: Yes

2. Has the statistical analysis been performed appropriately and rigorously?

Reviewer #1: I Don't Know

Reviewer #2: I Don't Know

3. Have the authors made all data underlying the findings in their manuscript fully available?

Reviewer #1: No

Reviewer #2: Yes

Response to Reviewer #1: We have now included a Data Availability Statement as follows:

“Our data are available upon request. Although data are pseudonymised, details such as the combination of sex, age, diagnoses, and period of assessment at ULMC and MTL could potentially be used to identify individual cases and patients. Therefore, in accordance with the General Data Protection Regulation, we are restrained regarding the public release of our data set. When administrative and legal requirements are met, trusted research institutions may request data access through the current director of the Institute of Laboratory Medicine, Clinical Chemistry and Molecular Diagnostics Leipzig (MB-sek-ilm@medizin.uni-leipzig.de) or the AMPEL project directly (MB-ilm-ampel@medizin.uni-leipzig.de).”

4. Is the manuscript presented in an intelligible fashion and written in standard English?

Reviewer #1: Yes

Reviewer #2: Yes

Response to the reviewer #1

1.) Introduction: 

The rationale for the study is outlined well. While it is more of a stylistic comment, it would be clearer to the reader if there was a separate paragraph for study aims and if they were enumerated.

- Response: Thank you very much for this suggestion. We have inserted a new paragraph of our aims and objectives as suggested (page 3, line 57).

2.) Methods:

2.1) Were kidney transplant recipients included or excluded from the study. This has not been stated.

- Response: This information is indeed missing. We have amended the methods section with the following statement (page 4, line 84-85). 

“Cases with kidney transplantation in patients’ history were not excluded, unless the diagnosis “N18.5 (dialysis-dependent chronic kidney disease)” was also coded.”

2.2) KDIGO classification / definition of AKI relies on knowing baseline serum creatinine. How were baseline serum creatinine levels ascertained?

- Response: Thank you for raising this point, which is a crucial one in the definition of AKI. The KDIGO guidelines state several possible routes for the determination of baseline sCr. One is the use of a standardized hypothetical eGFR of 75ml/min per 1.73m² and back-calculation for the specific age, sex and race. This only applies provided there is no evidence of CKD. Another is the assumption of a baseline as the lowest sCr during hospitalization. Point 2.1.1 of the KDIGO guidelines further specifies this sCr should have occurred within the last 7 days. Our CDSS is set out to apply the definition of AKI restrictively, rather than too loosely, in an attempt to limit the number of false alarms. Therefore, we refrained from using hypothetical sCr and used the following definition: Minimum sCr of a patient during hospitalization within the last 7 days. We amended the methods section to clear up this point (page 5, line 107-108):

“AKI was defined and staged based on absolute and relative changes of creatinine levels within the case, according to KDIGO-criteria, with the lowest sCr level of the last 168 hours serving as a baseline in this definition (Table 1).”

2.3) KDIGO classification also establishes AKI diagnosis based on urine output. This study does not include urine assessment. This is correctly stated in the limitation and should be made clear in the methods as well.

- Response: Good point, thank you! Now it is described in the methods section (page 5, line 109).

“AKI was defined and staged based on absolute and relative changes of creatinine levels within the case, according to KDIGO-criteria (Table 1). Urine excretion as an AKI criterion was not available.”

2.4) What was done for the same patient’s multiple admissions? This is unclear. If there are repeat admissions (as they are bound to be in healthcare systems), are all admissions included or only the first admission included? This is not clear.

- Response: We included all admissions in our analysis, which also means all repeated admissions of patients. Thank you for bringing up the point. We have now included a clear statement (page 4, line 74).

“In this retrospective observational cohort study, all hospitalized cases (which means all hospital admissions) (n = 227 194) admitted between 1st January 2017 and 31st December 2019 at the University of Leipzig Medical Center (ULMC, 1 451 beds) and the Muldentalkliniken (MTL, hospital locations Grimma (177 beds) and Wurzen (178 beds) were analysed, including repeated hospital admissions.”

2.5) It is also unclear if patients were transferred from primary hospitals to tertiary hospitals. If they are transferred, are they counted twice? Was this analyzed?

- Response: This raises an interesting point. Due to independent data pseudonymization at the analyzed primary and tertiary hospitals, we are not able to link data between hospitals and analyze patient transfers that might have occurred.

3.) Results:

3.1) It is not very surprising that tertiary care center was found to have a higher burden of AKI. From Table 2, it appears that MTL has older patients with lower baseline GFR. So, it is little surprising that despite having 11% AKI incidence, there were lower need for dialysis in MTL. This makes me wonder if patients were transferred to the tertiary hospital if dialysis was needed and that (if true) is not accounted for in the analysis or in the interpretation of data.

- Response: Thank you very much for this comment. In the discussion we go further into this topic. There we discuss several aspects that could be the reason for the different results between primary and tertiary hospitals (page 13, line 223 – 232). We did not conduct a direct analysis of patient transfers. We have included this point in the limitations (page 16, line 275-281).

“Thus, pre-stationary diagnostics and long-term outcome data, such as outpatient deaths, were not available. An analysis of patient transfers from MTL to a tertiary care hospital was not performed.”

3.2) As patients with AKI have a higher co-morbidity burden, it is unclear if AKI is an independent predictor of mortality or a marker of poor prognosis in this dataset (relating to Table 3). Multivariable analysis would be helpful. Dataset has enough event rate for multivariable analysis testing AKI as an independent predictor for mortality.

- Response: Thank you very much for bringing up this idea. After thorough reflection, we decided against a multivariable analysis testing AKI as an independent predictor for mortality in the context of our study. The study was not specifically designed to prove AKI as an independent risk factor. A case control study would be a more suitable setup for this objective. Furthermore, our data are based on routine patient treatment and documentation. Accordingly, relevant confounders were not systematically documented and cannot be controlled for. The different documentation quality of different wards would also result in a bias. Moreover, with a larger number of possible confounders, the number of cases is too small. We are therefore extremely hesitant to attempt the identification of causal factors.

3.3) Regarding the association between progressive AKI and clinical outcomes, does the site (ULMC vs. MTL) alone account for the differences in clinical outcomes related to age? ULMC patients were younger and, since it is a tertiary care center, likely sicker than MTL patients who were on average 13 year older. It would be helpful if this data was divided by sites (ULMC and MTL) to know if the relationship between progressive AKI and younger age still holds independently.

- Response: This is an interesting question, which we are currently exploring in depth in a follow-up study. The differences regarding clinical outcomes and baseline characteristics between progressive and non-progressive AKI persists in the current study even if ULMC or MTL are considered independently. A more detailed comparative analysis of patient characteristics of progressive and non-progressive AKI at the primary hospital is currently in preparation.

4.) Supplementary information - I am not sure if Plos One would require underlying data behind summary statistics to be provided as a data table. Summary statistics are well highlighted well but I did not see the underlying data points behind summary statistics.

- Response: We have not included the underlying data points behind summary statistics for reasons stated in the newly added “Data Availability Statement”. However, all data will be made available by any of the coauthors upon reasonable request. 

“Due to potentially identifying patient information in the data set such as the combination of sex, age, diagnoses and period of assessment at ULMC and MTL, data underlying the analyses of the current study will be made available by any of the coauthors upon request only.”

Response to the reviewer #2

Reviewer #2: Authors have showed the utility of AMPLE-CDSS with progression of AKI during hospitalization in their retrospective data collection among the defined three hospitals. This study is in the direction of our understanding whether this tool will ultimately help in improving AKI and patient outcomes. Well executed and well summarized data collection.

---

## [Editor Report · Decision Letter 1]

30 Jun 2021

Acute Kidney Injury and its Progression in Hospitalized Patients - Results from a Retrospective Multicentre Cohort Study with a Digital Decision Support System

PONE-D-21-07899R1

Dear Dr. Kister,

We’re pleased to inform you that your manuscript has been judged scientifically suitable for publication and will be formally accepted for publication once it meets all outstanding technical requirements.

Kind regards,

Bhagwan Dass, MD

Academic Editor

PLOS ONE
---

## [Editor Report · Acceptance letter]

2 Jul 2021

PONE-D-21-07899R1 

Acute Kidney Injury and its Progression in Hospitalized Patients - Results from a Retrospective Multicentre Cohort Study with a Digital Decision Support System 

Dear Dr. Kister:

I'm pleased to inform you that your manuscript has been deemed suitable for publication in PLOS ONE. Congratulations! Your manuscript is now with our production department. 

Kind regards, 

on behalf of

Dr. Bhagwan Dass 

Academic Editor

PLOS ONE